# Novel Descriptors and Digital Signal Processing- Based Method for Protein Sequence Activity Relationship Study

**DOI:** 10.3390/ijms20225640

**Published:** 2019-11-11

**Authors:** Nicolas T. Fontaine, Xavier F. Cadet, Iyanar Vetrivel

**Affiliations:** PEACCEL, Protein Engineering ACCELerator, 6 Square Albin Cachot, BOX 42, 75013 Paris, France; xavier.cadet.fjf@gmail.com (X.F.C.); iyanarvetrivel@gmail.com (I.V.)

**Keywords:** innov’SAR, artificial intelligence, machine learning, protein spectrum, rational screening, digital signal processing, extended sequence, directed evolution

## Abstract

The work aiming to unravel the correlation between protein sequence and function in the absence of structural information can be highly rewarding. We present a new way of considering descriptors from the amino acids index database for modeling and predicting the fitness value of a polypeptide chain. This approach includes the following steps: (i) Calculating Q elementary numerical sequences (Ele_SEQ) depending on the encoding of the amino acid residues, (ii) determining an extended numerical sequence (Ext_SEQ) by concatenating the Q elementary numerical sequences, wherein at least one elementary numerical sequence is a protein spectrum obtained by applying fast Fourier transformation (FFT), and (iii) predicting a value of fitness for polypeptide variants (train and/or validation set). These new descriptors were tested on four sets of proteins of different lengths (GLP-2, TNF alpha, cytochrome P450, and epoxide hydrolase) and activities (cAMP activation, binding affinity, thermostability and enantioselectivity). We show that the use of multiple physicochemical descriptors coupled with the implementation of the FFT, taking into account the interactions between residues of amino acids within the protein sequence, could lead to very significant improvement in the quality of models and predictions. The choice of the descriptor or of the combination of descriptors and/or FFT is dependent on the couple protein/fitness. This approach can provide potential users with value added to existing mutant libraries where screening efforts have so far been unsuccessful in finding improved polypeptide mutants for useful applications.

## 1. Introduction

Proteins are biological macromolecules consisting of one or more amino acid chains. Proteins differ from one another primarily in their sequence of amino acids; differences between sequences are called “mutations”. One of the ultimate goals of protein engineering is the design and construction of peptide, enzyme, protein, or amino acid sequences with desired properties (collectively called “fitness”). The construction of chimeric proteins (i.e., “mutants”) can be achieved by substitution, deletion or insertion of amino acids, or blocks of amino acids. This allows an assessment of the role of any particular amino acid in the fitness, and hence helps us understand the relationships between the protein structure and its fitness.

The main objective of the quantitative structure–function/fitness relationship analysis is to investigate and mathematically describe the effect of changes in the structure of a protein on its fitness. The impact of mutations is related to physicochemical and other molecular properties of amino acids which vary and can be assessed by means of statistical analysis. 

Mixed in vitro and in silico approaches have been developed to assist the process of directed evolution of proteins. They require that the wet laboratory construct a library of mutants by site-directed, random, or combinatorial mutagenesis. From this library, sequences and/or structures of a limited number of samples are retrieved to form the “learning data set”. The fitness of this set is assessed. On the in silico side, numerical descriptors for each mutant are extracted. Subsequently, in the “learning phase”, multivariate statistical method(s) are used to establish the relationship between the numerical descriptors and fitness. Finally, a model is established to make predictions of fitness for mutants which are not experimentally tested.

Methods requiring 3D structural information, called quantitative structure–function relationships (QSFR), were proposed by different authors [1,2,3,4,5,6]. The QSFR approach is efficient and takes into account information about possible interactions with non-variants residues. However, QSFR is limited by the requirement of a 3D protein structure, which is still limited at present. Other methods, which only require sequence information to perform in silico rational screening using statistical modeling, were also proposed [5,7,8,9,10,11,12,13]. Comparatively, they do not need knowledge of 3D structure, as they are computed based on primary sequence only, and they can use linear and non-linear models. However, it has been shown that the computing time may be the major limiting factor especially when the number of interaction terms to take into account is high [11,14] or the number of k-mers to consider is too large [4].

Digital signal processing (DSP) techniques are analytic procedures, which decompose and process signals in order to reveal information embedded in them [15,16]. The signals may be continuous or discrete, as is the case for protein residues. DSP such as Fourier transform have helped analyze protein interactions [17] and made biological functionalities calculable [18,19]. These studies have been reviewed in detail in Nwankwo N. and Seker H. (2011) [20]. Jia et al. (2015) [21] proposed DSP based on wavelet transform. Recently, we proposed a versatile and fast in silico approach, named innov’SAR, to help in the process of directed evolution of proteins based on DSP [22]. Our method allows the prediction of at least one fitness value of a protein based on a protein spectrum obtained after applying fast Fourier transform (FFT) to a numerically encoded amino acid sequence. The approach outperformed the other known methods. We also demonstrated that this approach can be applied not only to predict additive effects but also epistatic interactions between amino acid residues, as is the case for predicting the enantioselectivity of epoxide hydrolase [23]. Innov’SAR is also efficient for multiple enzyme parameter determination, as exemplified by glucose oxidase, where enzyme activity, in the presence of different mediators across a range of pH values has been improved up to 121 times [24].

Improving the quality of prediction models is a constant challenge. Different types of descriptors have been tested for this purpose by different groups: Physicochemical properties and z-scales [7,8], binary encodings [9,10,25] with or without considering the interactions between mutations. Recently, Barley et al. (2018) [26] have proposed improved descriptors for the quantitative structure-activity relationship modeling of peptides and proteins by changing the hydrophilicity scale suggested by Hellberg [7].

In our recent articles [22,23,24] we selected and used for modeling the best descriptor chosen from the ensemble of descriptors listed in the AAindex database [27]. Each amino acid index is based on one or more physicochemical properties such as hydrophobicity, alpha and turn propensities, or others. Each index gives specific information about the protein. Our working hypothesis here is that the use of several descriptors at the same time should provide more useful information about the protein as modeling input. Our goal is to see whether it is possible to improve the results of the predictions by (i) the use of multiple physicochemical descriptors, (ii) the combination of multiple descriptors coupled with the implementation of FFT, and (iii) by classification of the descriptors based on general protein features. We wanted to implement a method that could be run in an acceptable time span with limited resources, i.e., typically on a laptop. Consequently, we limited the number of descriptors used for modeling. Two approaches were explored to find the best set of descriptors to use for a specific dataset. The comparisons are illustrated on four datasets that are available in the public domain: Cytochrome P450 for thermostability, Tumor necrosis factor (TNF) alpha for binding affinity, Glucagon-like peptide-2 (GLP-2) for potency, and epoxide hydrolase for enantioselectivity. The choice of the datasets was made: (i) To show that the choice of the combination of descriptors and/or the implementation of FFT is dependent on the protein and the fitness we are trying to predict, and (ii) to illustrate the ability of the approach to perform when limited training data is available.

In some cases, this approach allows a significant improvement in the quality of models and fitness predictions. The implementation of FFT taking into consideration the interactions between amino amide residues leads to the best predictive models.

The study is organized as follows. Section 4 presents the experimental datasets and the modeling approach. Section 3 describes the results obtained, and Section 2 discuss these results. The concluding remarks are given in Section 5.

## 2. Results

A protein sequence could be represented in the form of various numerical sequences, derived from different encodings. An elementary numerical sequence is one of these different forms of numerical sequences.

The impact of the use of several indices for the encoding of elementary sequences has been evaluated. Similarly, we tested the effect of different ways of combining elementary sequences to obtain extended sequences:

By successive concatenation after selection of the best indices, by combinatorics of the best indices, the elementary sequences having been subjected or not to a Fourier transformation.

Moreover, given that accumulation of elementary sequences leads to an increase in computation time, we examined the impact of the selection of a limited number of descriptors by selecting 20% of the descriptors of the extended sequences, in order to shorten the calculation time.

### 2.1. Concatenation to Obtain Extended Numerical Sequences

At the end of the encoding step, as described in the “*Materials & Methods*” Section 4: (1), All the protein variants from the initial dataset are encoded numerically according to an index from the AAindex database. This kind of encoded sequence is called “*noFFT_Seq*”; (2), if FFT is applied, each sequence will be transformed into a protein spectrum. This kind of encoded sequence is called “*FFT_Seq*”.

A sequence encoded by one given index, with or without FFT, is called an elementary sequence (Ele_SEQ). In our previous papers [23,24], only Ele_SEQ were used as input signals for the modeling step.

An extended numerical sequence (Ext_SEQ) is obtained by concatenating Q Ele_SEQ, where Q ≥ 2. Q is the number of Ele_SEQ used for sequence encoding. As mentioned above, in our previous works all the encodings were done with Q = 1. In an Ext_SEQ, all the Ele_SEQ are distinct from each other. 

The 566 indices in the AAindex database can be used to encode the numerical sequence with (denoted *FFT_Seq*) or without (denoted *noFFT_Seq*) the application of Fourier transform, hence producing 1132 unique elementary sequences that can be incorporated in the Ext_SEQ. When multiple encoding indices (denoted j1, j2…j566) are considered with or without applying Fourier transform, they are represented as FFT_Seq_j1_, FFT_Seq_j2_… and noFFT_Seq_j1_, noFFT_Seq_j2_…, respectively. 

As an example, if the amino acid sequence of the protein is encoded by only one index, the Ext_SEQ is obtained according to the following formulation:Ext_SEQ = noFFT_Seq--FFT_Seq(1)
where the symbol “--” between the two Ele_SEQnoFFT_Seq and FFT_Seq represents the concatenation of these two Ele_SEQ.

As another example, if the amino acid sequence of the protein is encoded according to two distinct encoding indices j1 and j2, the Ext_SEQ is represented according to the following possible alternative formulations:Ext_SEQ = noFFT_Seq_j1_--noFFT_Seq_j2_(2)
Ext_SEQ = FFT_Seq_j1_--noFFT_Seq_j2_(3)
Ext_SEQ = noFFT_Seq_j1_--FFT_Seq_j2_(4)
Ext_SEQ = FFT_Seq_j1_--FFT_Seq_j2_(5)

The order of the Ele_SEQ is not relevant for the construction of Ext_SEQ. An Ext_SEQ is a combination of Ele_SEQ. So, for example, the Ext_SEQ (5) could also be represented in the form FFT_Seq_j2_--FFT_Seq_j1_.

This general procedure is summed up in Figure 1.

We can derive from the above formulations the possible alternate variants of the Ext_SEQ in the case where the amino acid sequence of the protein is encoded according to a number Nb_Index of distinct encoding indices j1, j2, …, j_Nb_Index_ which is strictly greater than two.

In other words, the concatenation pattern defines, for each elementary numerical sequence in the Ext_SEQ, the respective index, and whether or not the Fourier transform was applied. In the Ext_SEQ obtained by concatenating Q number of Ele_SEQ, all the Ele_SEQ are distinct from each other. For example, given a pair of Ele_SEQ, they could differ either in the AAindex encoding them or the utilization of FFT, or both. So, the extended numerical sequence Ext_SEQ is determined according to the formulation (1) in the case of a single encoding index, or according to any one of the formulations (2) to (5) in the case of two distinct encoding indices, or according to similar formulations with at least Ele_SEQ elementary numerical sequences in the case of more than two distinct encoding indices.

In our modeling method described in “*Materials & Methods*”, Ele_SEQ and the experimental values of the target activity are the standard inputs for encoded sequences. However, our modeling method could also accept Ext_SEQ instead of Ele_SEQ as input in order to generate predictive models.

In this study, we generated Ext_SEQ with a maximum of *Q* = 3 Ele_SEQ to avoid the construction of large encoded inputs and to ensure modeling runs with limited computational resources.

### 2.2. Combining One or Multiple Indices

As explained above, two types of representation are considered for an index: *noFFT_Seq* and *FFT_Seq.* A protein sequence can be encoded in different ways:

Using a single encoding index or using multiple encoding indices,

combining “*noFFT_Seq*” and/or “*FFT_Seq*” for the single index or the combination of indices.

These two cases are described and tested below.

#### 2.2.1. Concatenation of the Best Single Indices with or/and without FFT

We have shown that the Fourier transformation into a protein spectrum has a significant impact on the predictive ability of the models. Indeed, if we consider the TNF dataset, using a leave-one-out cross-validation (LOOCV) procedure, the coefficient of determination in cross validation (cvR^2^) drops from 0.85 with FFT to 0.64 without FFT and the root mean squared error in cross validation (cvRMSE) rises from 0.32 to 0.48, using only the best single index: Thus, applying FFT significantly improves the quality of the model [22].

In this study, we determined the best single index for modeling with FFT and the best single index for modeling without FFT. We wanted to see if the use of these indices, as a concatenation, could improve the performance.

First, we evaluated whether the combination of two Ele_SEQ derived from the same index, one with FFT and one without FFT, could improve modeling performances. Figure 2a shows the results obtained for cytochrome P450 using the best index obtained when FFT is applied (FFT_Seq): CvR^2^ and cvRMSE are respectively 0.83 and 1.91. The results obtained using this best index when an extended sequence noFFT_i1_-FFT_i1_ generated are cvR^2^ = 0.83 and cvRMSE = 1.89 (Figure 2b). The *p*-value associated with Student’s t-test used to assess if the difference for the quadratic errors between the two models indicates that this slight decrease in cvRMSE is not significant (*p*-value = 0.7781).

Next, we evaluated whether the combination of the best single index with FFT and the best single index without FFT could lead to more interesting results. Thus, these indices were used to get an extended sequence “Ext_SEQ” such as noFFT_Seq_i1_-FFT_Seq _i2_ (Figure 2c) for cytochrome P450. Here, in parallel, the best index was selected without FFT. Figure 2c shows the results obtained for the extended sequence noFFT_i2_-FFTi_1:_ cvR^2^ and cvRMSE to be respectively 0.84 and 1.85. *p*-value decreases but remains quite high in this case (*p*-value = 0.4123).

#### 2.2.2. Combinatorial of a Protein Sequence Encoded by Multiple Indices with FFT

We explored the concatenation of multiple indices with FFT. The first idea to explore this approach on a given dataset is to generate all the possible Ext_SEQ produced from the combination of all available Ele_SEQ. The formula to obtain the number of combinations is:(6)$$\overset{Q=N}{\underset{Q=1}{\sum }}\frac{N!}{Q!\left(N-Q\right)!}$$

In our case, *N* is the number of indices and *Q* is the number of used Ele_SEQ for the generation of Ext_Seq.

However, with *N* = 566 indices from the AAindex database, this ensemble of Ext_SEQ represents more of 2.4e + 170 possible encoding inputs for the modeling. In order to limit this number, we tested two methods to screen a smaller number of combinations of Ele_SEQ and to find appropriate Ext_SEQ for the modeling performance: (i) Using a combinatorial approach or (ii) by successive concatenation, based on the selection of the best indices.

##### Combinatorial Sequences Approach by Using a Selection of Best Single Indices

This first approach consists of restraining the number of indices used for the encoding in order to have a smaller ensemble of Ext_SEQ. We call it “*one phase combinatorial approach*” because the idea behind it is to generate all the possible combinations of Ele_SEQ from a limited selected list of indices in one phase.

First, a modeling is done on the dataset with an encoding using each of the possible 566 AAindices, one by one and without indices concatenation: i.e., only Ele_SEQ, and not Ext_SEQ, are used as modeling inputs. The model performances are ranked, based on the lowest cvRMSE, to identify the best index to use alone. The ranking allows identification of the N best indices to use alone and which have to be selected for the combination of Ele_SEQ, so as to generate Ext_SEQ. The modulation of N allows modifying the size of our ensemble of Ext_SEQ.

At this stage, all the Ext_SEQ are built in one phase. In our study, we limited N to the top 10 best indices and *Q* ≤ 3. Thus, one sequence could be represented by 175 Ext_SEQ. Next, each Ext_SEQ is used as modeling input and a ranking of the models is performed. The best Ext_SEQ can thus be identified.

To illustrate this approach, let us consider the top 10 indices obtained after a ranking of the 566 indices, and let us consider only the “FFT_Seq”. The outcome after the combinatorial process could be, for example:

FFT_Seq_j1_--FFT_Seq_ji2_

FFT_Seq_j2_--FFT_Seq _j3_

FFT_Seq_j1_--FFT_Seq _j2_^_^FFT_Seq_j4_

FFT_Seq_j1_-- FFT_Seq_j2_--FFT_Seq_j3_--FFT_Seq_j4_--“FFT_Seq_j5_--FFT_Seq_j6_-- FFT_Seq_j7_--FFT_Seq_j8_--FFT_Seq_j9_-- FFT_Seq_j10_

This “*one phase combinatorial approach*” is applied to our four datasets (GLP-2, TNF alpha, cytochrome P450, and epoxide hydrolase) in order to evaluate and identify the better models.

***GLP-2***. We applied this approach to the GLP-2 dataset and from a previous study [22] the index 449 was shown as the best after a ranking of indices and encoding with FFT. 

Figure 3 shows the results obtained with the index 449 alone and its model identified in our previous work [22]. CvR^2^ and cvRMSE are respectively 0.42 and 2.11 for the index 449.

We used the ranking of the indices and applied the *one phase combinatorial approach*. A combinatorial of three indices at most is run on the top 10 indices from the previous ranking. As FFT_Seqj_1_-FFT_Seqj_2_ is equivalent to FFT_Seqi_2_-FFT_Seq_i_1_, 175 combined extended sequences are obtained.

Table 1 shows that the best obtained cvR^2^ and cvRMSE with three indices are respectively 0.47 and 1.99, with *p*-value = 0.53. Thus, in this case the modeling performance appears better but the improvement is not significant according to the *p*-value (*p*-value = 0.531). Nevertheless, interesting findings were obtained. Indeed, we tested the model using the 10 indices to form the Ext_SEQ FFT_i1_-FFT_i2_….FFT_i10._ The cvRMSE of this model jumps to 2.48 and the cvR^2^ decreases to 0.11. So, it should be noted that the right number of indices has to be found: i.e., a combination of *m* index is not always better than a combination of *n* index (with m > n). In other words, large Ext_SEQ is not equal to better modeling performances. The addition of indices for the encoding step is not related to the improvement of the modeling. Moreover, we notice that index 449, the best index when only one index is selected, could not be the best to use for a combination of three indices from the top ten indices as exemplified in Table 1. Indeed, 449 alone appears in position nine in the ranking.


***Epoxide Hydrolase***


We performed the same operation, with application of FFT, on the epoxide hydrolase dataset and results are presented in Table 2.

In another study [23] we identified, for epoxide hydrolase, the index 303 as the best after a ranking of indices. Figure 4 shows the results obtained with the model based on the index 303 alone.

The performance with the best index, index 303, was already high, 0.96 and 0.12 for cvRMSE and cvR^2^, respectively. The best performances, seen in Table 2, are 0.105 and 0.969, respectively for cvRMSE and cvR^2^, with the *p*-value = 0.43. Thus, the combinatorial of multiple indices appears to slightly improve the modeling performances but the improvement is not significant.

It should be noted that index 303, identified as the best when ranking the 566 indices, is classified only in position 38 (Top 38) when a combinatorial approach is used: i.e., 37 combinations of indices are better than 303 alone (when considering only the top 10 in this example and when this best index (i.e., 303 here) is included in the top 10).

On the other datasets, we also did not have significant improvement with the combinatorial approach. One reason could be that the top 10 selected indices were not the best for the concatenation with three indices. So, we implemented another method with a different way of selecting an index for the improvement of modeling performances.

##### Successive Concatenation of a Protein Sequence Encoded by Multiple Indices

The second method is termed “*successive concatenation*” because we increase the size of Ext_SEQ incrementally in several iterations. In “*successive concatenation*”, N iterations of innov’SAR modeling are applied to find the best N indices and the associated Ext_SEQ, in an iterative process.

For each iteration, the best previous index or indices is/are kept. This allows us to construct incrementally the Ext_SEQ with different indices, i.e., at the end of an iteration, the best index for the modeling performances is determined and it will be kept for the next iteration.

In the first iteration, with the AAindex including 566 indices, the 566 indices are evaluated one by one as described in the modeling approach based on one index. A ranking of the 566 indices is performed according to cvRMSE values (from the cross-validation procedure) as detailed in Section “*Modeling approach*”. The best index j1 is the one that gives the lowest cvRMSE. Consequently, the index j1 is the first index used to construct the first part of the Ext_SEQ. The protein sequence is encoded according to the first index j_1_, using a sequence representation *noFFT_Seq* or *FFT_Seq*. In the second iteration, the process identified another index, j2, to use for the construction of Ext_SEQ of two Ele_SEQ, starting from the sequence encoded by j1 as a base block of Ext_SEQ. The index j2 is identified by a second ranking with all the indices except the one used in the base block of Ext_SEQ, j1, for the second iteration, i.e., the ranking on 566—one indice. For each iteration the same operation is repeated to find the best index for modeling and increasing the size of the Ext_SEQ.

Thus, an extended sequence Ext_SEQ such as FFT_Seq_j1_-FFT_Seq _j2_-..-FFT_Seq_jn_ is obtained. This could be extended to any number of parts in the “Ext_SEQ”. Furthermore, a mix of noFFT and FFT could be used.

This procedure is illustrated in Figure 5.

Exemplification of this procedure is done with the four datasets with three indices.


***GLP-2 Dataset***


The obtained modeling performances with the first best single index, 449, are 0.42 and 2.11, for cvR^2^ and cvRMSE, respectively (cf. Figure 3). Figure 6 shows the results obtained using the three indices, 449, 341, and 193, gathered in the Ext_SEQ “FFT_SEQ_j1_--FFT_Seq_j2_--FFT_Seq_j3_”. cvR^2^ and cvRMSE are 0.55 and 1.75, respectively. Thus, using the three indices significantly improves the quality of the prediction. This is confirmed by the *p*-value equal to 0.008 in Student’s test for the significance of the improvement.


***Epoxide Hydrolase Dataset***


We showed the best model based on one index in Figure 4 with 0.96 and 0.12, respectively for the cvR^2^ and the cvRMSE. Figure 7 resulted from the application of the successive concatenation method to the epoxide hydrolase dataset. The combination with the indices 14 and 234 gives better performances since 0.97 and 0.09 are respectively obtained for the cvR^2^ and the cvRMSE but the improvement in comparison to one index is not significant, with a *p*-value of 0.343. Nevertheless, we note that here the *p*-value is lower than the value (0.43) shown in Table 2 above.


***TNF Alpha Dataset***


The performances with the best index (Figure 8a), index 203, are 0.85 and 0.32, respectively for the cvR^2^ and cvRMSE, for the TNF dataset. The combination with the indices 504 and 486 (Figure 8b) allows increasing cvR^2^ to 0.88 and decreasing cvRMSE to 0.28 (*p*-value = 0.175).


***Cytochrome P450 Dataset***


The same method was applied to the cytochrome P450 dataset and is shown in Figure 9. The performances with the best index, index 300, are shown in Figure 2a, with 0.83 as cvR^2^ and 1.91 as cvRMSE. The combination with the indices 39 and 226 allows significant improvement of these performances, up to 0.88 and 1.63, respectively for the cvR^2^ and the cvRMSE. This improvement is confirmed by the *p*-value (0.002).

### 2.3. Selection of Indices from Different Families for Concatenation or Combination

We saw above that when the best indices are selected successively or when running a combinatorial, an improvement of the prediction could be obtained. As we stated in our previous work [23], the selection of the best index by innov’SAR in a round of modeling is done using a statistical approach. Innov’SAR selects an index without the comprehension of the protein feature associated with this index. Consequently, in all our rounds to extend the encoded sequence, the best chosen Ele_SEQ are selected in the statistical approach for their ability to improve the modeling performance.

One may ask if a selection of the best indices based on their biochemical properties and protein features would not lead to further improvement. Indeed, by using indices from different families, the encoding input will allow describing the proteins with more biochemical features and will give greater scope for the modeling to be analyzed.

We decided to analyze the protein feature of each index and to find out if the use of several biochemical features improves the performances. We also tried to determine if we can rationalize, in a biological way, the construction of Ext_SEQ, in order to obtain better modeling performances.

Four hundred and two amino acid indices of the AAindex database were classified into the following families of general protein features by Tomii and Kanehisa [28]:Alpha and turn propensities,beta propensity,composition, hydrophobicity,physicochemical properties, other properties.

For the other 164 indices not analyzed by Tomii and Kanehisa [28], we cannot associate them with one of the six families of protein features. We placed them into a seventh arbitrary family of non- assigned features, labeled NA. We used this classification to associate a general protein feature with an encoded input. Each Ele_SEQ of an Ext_SEQ can be associated with a described family. We used the successive concatenation approach to construct Ext_SEQ. After the generation of these inputs, we determined the family of protein feature for each Ele_SEQ included inside the Ext_SEQ as seen in Table 3.

For the GLP-2 dataset, the identified model by the successive concatenation, whose performances are shown in Figure 5, is based on a combination of three indices from three different families: Index 449 associated with the other properties families, index 341 associated with the alpha and turn propensities, index 193 associated with the composition. Thus, compared to the model based on one index and one protein feature (Figure 3), a combination of indices from different families could significantly improve the results, in this case (*p*-value = 0.0078).

For the epoxide dataset, the best model was also based on three different families of protein features: Other properties, hydrophobicity, beta propensity. However, the improvement was not significant. 

For the TNF alpha dataset, the model linked to Figure 8b is based on the index 203 associated with the composition family. However, we cannot identify the family protein feature for the other two indices used in the model.

In the cytochrome datasets, the model linked to Figure 9 was associated with two families of indices: With index 300 attached to the alpha and turn propensities family and indices 39 and 226 attached to the beta propensity family. Thus, with our successive concatenation approach, it is not relevant to use the maximum protein feature families to improve the modeling performance. Therefore, we can also deduce that an Ext_SEQ with *Q* = 6 and using the six distinct families is not bound to be the best Ext_SEQ for the improvement of modeling performances.

The next question is linked to the number of variables to handle during the modeling; we investigate the impact of a selection of variables as input data.

### 2.4. Selection of Variables Inside the Ext_SEQ

A representation of a protein spectrum is shown as a plot: Energy = f(frequency) [22]. It might be useful to limit the number of frequencies of the protein spectrum in the interest of saving both computational power and time. This number varies according to the protein studied: Thirty-three for GLP-2, 129 for TNF, 257 for both cytochrome P450 and epoxide hydrolase. One possible method of reducing the number of frequencies is to select the most representative ones.

Our approach is applicable to the entire protein sequence, as shown in the previous examples, or to a selection of positions in the protein sequence without FFT and/or to a selection of frequencies in the protein spectrum of the FFT. Here, the selection of positions is performed in a similar manner as the selection of frequencies (or harmonics), i.e., by using a filter method.

A filter method selects variables regardless of the model and is, for example, based only on the correlation with the variable to predict. A filter method suppresses the least interesting variables. The other variables will be part of a classification or a regression model used to classify or to predict the class or the activity/fitness. Such a method is carried out, for example, by correlating amplitude values at each frequency with activity values (i.e., the values to be predicted), and then for selecting the frequencies with the highest correlation. The correlation is evaluated according to the R^2^, and the set of frequencies is then given a percentage of frequencies for which R^2^ is the highest. A given set of frequencies was selected and used for modeling.

Figure 10a–d illustrate this procedure, wherein the prediction method is carried out for a selection of frequencies in the protein spectrum of the FFT, i.e., for a given set of frequencies. In this example, the amino acid sequence of cytochrome P450 was encoded into a numerical sequence using a single best encoding index (index number 300) for Figure 10a,b, and using the two best encoding indices (index numbers 300 and 343) for Figure 10c. 

Figure 10a represents a plot of measured thermostability of cytochrome P450 variants versus thermostability, using the encoding index with number 300, while applying a fast Fourier transform to the encoded numerical sequence, but only for a given set of frequencies representing a part of the whole spectrum. In this example, the set of frequencies represents 20% of the whole considered spectrum. The frequencies are numbered from zero to 256 and the selected frequencies in this example are the following: 3; 7; 18; 22; 29; 33; 42; 46; 48; 58; 59; 65; 69; 79; 81; 88; 94; 99; 103; 109; 111; 112; 115; 128; 132; 134; 138; 139; 142;146; 159; 160; 163; 165; 171; 177; 182; 183; 184; 206; 214; 220; 222; 223; 224; 225; 226; 230; 235; 238; 240; 249. Therefore, Figure 10a corresponds to FFT_20%_.

Figure 10b represents a similar plot with index number 300 for two elementary numerical sequences, one elementary numerical sequence without FFT and the other with FFT, but only for a given set of frequencies or harmonics representing 20% of the whole spectrum. Thus, Figure 10b corresponds to the extended numerical sequence Ext_SEQ equal to noFFT_Seq-- FFT_20%__Seq with index number 300.

Figure 10c represents a plot with the two best encoding indices (index numbers 300 and 343) for two elementary numerical sequences, one (index number 343) without further applying Fourier transform to the elementary numerical sequence, and the other (index number 300) with further application of the fast Fourier transform, but only for a given set of frequencies representing 20% of the whole spectrum. Figure 10c therefore corresponds to the extended numerical sequence Ext_SEQ equal to no FFT_Seq_j1_-- FFT_20%__Seq_j2_ with j1 equal to index number 343 and j2 equal to index number 300.

Figure 10a shows the results obtained with the best index 300 alone and FFT_20%_: CvR^2^ and cvRMSE are 0.66 and 2.68, respectively. With the same encoding index 300, without FFT and with FFT_20%_, Figure 10b shows better results obtained with the prediction method: CvR^2^ and cvRMSE are 0.74 and 2.38, respectively. With the two best encoding indices (index numbers 300 and 343) and FFT_20%_ for index number 300, Figure 10c shows better results obtained with the prediction method: cvR^2^ and cvRMSE are 0.74 and 2.39, respectively.

Figure 10d, using three indices, 300, 39, and 226, with FFT, with conservation of 20% frequencies. cvR^2^ and cvRMSE are 0.70 and 2.52, respectively. In each case, the improvement is significant compared to the reference (see Table 4).

Table 4, which sums up all the values of the performance parameters, shows the cvR^2^ and cvRMSE values for Figure 10a–d.

Based on the cytochrome P450 dataset, the following observations can be made:(i)Keeping all the frequencies when using one index leads to better cvR^2^ and cvRMSE. This statement is exemplified by Figure 1a, where cvR^2^ = 0.83, cvRMSE = 1.91; and Figure 10a, where cvR^2^ = 0.66, cvRMSE = 2.68.(ii)when combining multiple index and FFT, the predictions are better when using all the variables in the protein spectrum: CvR^2^ values are 0.70 and 0.88 and cvRMSE values are 2.52 and 1.63, respectively for Figure 9 and Figure 10d, respectively.

So, a reduction of the variables can be a good option to minimize the calculation time, but at the cost of the quality of the prediction. One hypothesis to explain this observation is that among the eliminated frequencies some contain part of the information relating to the interactions and that this information about interactions between the positions of the amino acids in the polypeptide sequence is important for establishing a solid predictive model.

## 3. Discussion

### 3.1. Cumulating Indices Could Provide Better Prediction Performances

The examples above show that when an accumulation of indices is used, better results are obtained in terms of cvRMSE and cvR^2^. Table 4 sums up the performance metrics in LOOCV.

The *p*-value associated with the Student’s t-test provides further information. It makes it possible to know if the observed improvement in terms of cvR^2^ and cvRMSE from one model to another is significant or not, i.e., to know if the accumulation of indices, with or without FFT, produces an improvement. Different cases are observed with respect to the *p*-value associated with the difference for the quadratic errors between two models; they are summarized in Table 4:

For cytochrome P450: When the FFT is not implemented, the models with Ext_SEQ are not significantly different from the reference model. On the other hand, when two or three indices are cumulative, and the FFT implemented, the models are significantly different: Indeed, for example, we pass from the reference model with cvR^2^ = 0.83 and cvRMSE = 1.91 to 0.88 and 1.63, respectively for Ext_SEQ with three indices. Student’s t-test shows a significant improvement of performances with a *p*-value = 1.95 × 10^−3^.

A similar finding is made for GLP-2: *p*-values obtained with Student’s t-test indicate that the accumulation of indices significantly improves the reference model. 

In the case of epoxide hydrolase, the *p*-value indicates that the probability that the difference for the quadratic errors between two models is at least the obtained value, 0.34, assuming equal means. It is not surprising, since the basic model is already excellent with one index, that the accumulation of indices does not bring much improvement. Nevertheless, it should be kept in mind that a slight improvement in the quality of an excellent model may be important when trying to predict epistatic effects. We have already shown that the reference model in the context of epoxide hydrolase captures the epistatic effects, and this experimental verification [23] has been done.

In the case of TNF alpha, Student’s t-test does not allow concluding that the improvement of the models is significant at 95%.

We used two approaches for the concatenation of indices. 

The “*successive concatenation*” showed better performances than the “*one phase combinatorial*” approach. In successive concatenation, we explored the modeling using Ext_SEQ with *Q* ≤ 3. However, we did not screen all the possible Ext_SEQ with *Q* ≤ 3. Indeed, the successive concatenation is also limited in the space of generated Ext_SEQ, because for each iteration the indices from the previous iteration are inevitably kept. An Ext_SEQ outside of this screening space could further improve the predictions.

Ideally, the combinatorial approach using all the indices would allow finding the best Ext_SEQ, but the space of Ext_SEQ to explore is difficult to perform on a laptop.

However, our combinatorial approach even with a restricted number of indices allows us to show that a large accumulation of Ele_SEQ does not result in better modeling performances. Using the combinatorial approach, we created (see Table 1) models using fewer indices with better performances than models with more indices. We also show that the 10 best indices for modeling with one index were not the best to use for the generation Ext_SEQ.

In both approaches, complementarity in the descriptors is used to improve the predictive power of the model.

### 3.2. PLSR Modeling Using FFT and Extended Sequence Leads to Better Results 

We have shown in a previous paper [22] that the Fourier transformation into protein spectrum brings a significant improvement to the predictive ability of the models. We used an enzyme, epoxide hydrolase, to show that a model based on protein spectrum takes into account the effect of interactions between the residues at variable positions [23]. 

This kind of representation (protein spectrum) is more informative to effectively bring to light correlations that are not apparent otherwise. Our assumption is that the effect of a single point mutation on protein fitness is not purely local, but globally distributed over the linear sequence of the protein. Indeed, a single point mutation impacts the entire *protein spectrum* [22]. The effect of the mutation on fitness is seen as a global phenomenon, not a local phenomenon, as in other methods applying protein sequence to an activity relationship [5,11].

The results above, and summed up in Table 4, clearly indicate that an extended numerical sequence Ext_SEQ that includes Fourier transformation leads to better models and more efficient prediction.

### 3.3. Considerations on the Nature of the Descriptors and Versatility of the Approach

Four datasets have been used to support the method described here: GLP-2, TNF alpha, cytochrome P450, and epoxide hydrolase. These are proteins of different sizes: 33, 157, 464–466, and 398 amino acids, respectively. The targeted activities in prediction are also different: cAMP activation, binding affinity, enantioselectivity, and thermostability, respectively.

Table 5 indicates, for each protein studied, the physicochemical properties of each of the descriptors. A first observation is that whatever the protein, there is no obvious logic as to the biological nature that underlies the choice of this or that descriptor. A second observation is that when indices are selected to be cumulated, depending on whether the FFT is implemented or not, the choice will not be the same: The index retained for a protein sequence that will simply be encoded (without FFT therefore) is not the same as it will be if an FFT is applied. A third observation is that the choice of descriptors selected in combinatorial descriptors (see “Combinatorial of multiple indices”) is not the same as that which will be made during a successive selection.

Another interesting observation is the following: Syntheses of several descriptors across z-scales or principal components have been tested by different authors [7,8,9,26,29]. These synthetic descriptors can be seen as the quintessence of different descriptors representing various and complementary physicochemical properties. For example, the amino acid descriptors scales z1, z2 and z3 represent the three scores vectors of a principal component analysis of a table with 20 properties for the 20 coded amino acids. z1 is mainly related to hydrophobicity, z2 contains additional information from the size and hydrophobicity/hydrophilicity scales, whiles z3 contains information from pKCOOH, pI, and 1H NMR variables. It might have been expected that these summaries of descriptors would be retained first, but that is not the case. It is observed, in Table 5, that the selected indices are different from the z-scales (indices number 390–392) and principal components (index number 359). 

Generating, as we have done, extended numerical sequences is a way to choose a set of descriptors, more informative than a single descriptor in terms of fitness, to make the prediction. This is another way of understanding multi-faceted sequences in terms of physicochemical properties. Indeed, as exemplified in Table 5, we can see a large diversity in terms of physicochemical characteristics addressed by the indices. This observation is reinforced by the identification of the protein feature families involved in modeling as seen in Table 3. The improvements obtained show the relevance of this approach.

By using the protein feature families described by Tomii et al. [28], we analyzed multi-faceted sequences to find if we can use these protein features to improve the performance. In the case GLP-2 and epoxide hydrolase, the best Ext_SEQ used three different families, for *Q* = 3. So, the use of different protein features in Ext_SEQ could improve the modeling performances. In the case of cytochrome P450, our best Ext_SEQ used two different families, for *Q* = 3. Too much accumulation of physicochemical information is not always the best solution in terms of performance. Sometimes we obtain better results by using only indices from a few families instead of the larger or the whole list of families. In the case of TNF alpha, we do not have the family for two indices; an update with the Tomii et al. approach would allow the identification of these indices and the others not already classified. However, we have to keep in mind that it is also not clear if all the biological activities existing in proteomics could be linked to one or several classes described by Kawashima and the AAindex database [27]. It could be interesting to set in place other classification approaches of the indices. These classifications could be made based on protein features such as Kawashima’s, but with other protein features or more complex features. It could also be done in a statistical approach based more on a clustering of the different indices by the evaluation of the intrinsic value for each amino acid and each index. In both cases, complex experiments are required to evaluate the pertinence of a new classification approach.

These considerations show the need for further study of the descriptors and their consequence for modeling. This will be the object of a future project of our team.

It can be observed that the approach based on Ext_SEQ is versatile in terms of efficiency in predicting different kinds of fitness. Indeed, we chose four different protein fitness (cAMP activation, binding affinity, enantioselectivity, and thermostability) and the prediction, as observed in Table 4, were good, with the cvR^2^ varying from 0.55 to 0.97 using the descriptor Ext_SEQ (with FFT). cAMP activation refers to the potency and to the measure of drug activity expressed in terms of the amount required to produce an effect of given intensity. Binding affinity refers to the strength of interactions between proteins or proteins and ligands (peptide or small chemical molecule). The thermostability is usually expressed in °C and usually refers to the measured activity T_50_ defined as the temperature at which 50% of the protein is irreversibly denatured after an incubation time of 10 min. Finally, enantioselectivity refers to the selectivity of a reaction towards one of a pair of enantiomers.

The term “fitness” of a protein refers to its adaptation to a criterion, such as catalytic efficacy, catalytic activity, kinetic constant, Km, Keq, binding affinity, thermostability, solubility, aggregation, potency, toxicity, allergenicity, immunogenicity, thermodynamic stability, flexibility, etc. So, obviously there are many more possible fitness measurements to be tested than the four targeted in this study. The difficulty mainly lies in the availability of the corresponding training data set which is the starting point to build a model. However, in our opinion, these four examples already demonstrate the versatility of the approach based on this new kind of descriptor.

One last word on the calculation time. We stated in the introduction that one of our objectives was to shorten the calculation time so as a personal computer could be used.

### 3.4. Calculation Time 

We measured the calculation time, on a laptop, of different combinations and types of modeling. All steps of the innov’SAR approach were implemented on a workstation equipped with four processors i7 of 2.6 GHz and 18 GB of RAM. Table 6 below gives the time for different datasets and for the two methods shown previously, the concatenation of indices in several rounds (A) and the combinatorial of multiple indices in the same round (B).

The objective of this table is to give the reader an idea of the calculation time, not to compare the time between methods. Indeed, method A does several rounds on the 566 indices at each round, while method B uses only the top 10 indices from the ranking with one index. Method B could be configured to use 566 indices, but the number of combinations and the calculation time would be greater. Table 6 shows that the calculation time remains reasonable in each situation.

## 4. Materials and Methods 

### 4.1. Datasets

#### 4.1.1. Epoxide Hydrolase

The epoxide hydrolase dataset used by Reetz et al. [45] is a collection of 37 mutants and one WT sequence from *Aspergillus niger* organisms and their enantioselectivity. This enzyme is known for the hydrolysis of glycidyl phenyl ether. The epoxide hydrolase allows the synthesis of important intermediates for the synthesis of beta-blockers, commonly used pharmaceutical drugs in hypertension treatment [46]. Enantioselectivity refers to the selectivity of a reaction towards one enantiomer. The study by Reetz et al. [45] identified epoxide hydrolase mutants with an improved selectivity toward the S enantiomer.

Enantioselectivity is expressed by the E-value with a range between 0 and 115. The E-value can be transformed into ΔΔG^‡^ (kcal/mol) by the relation $\Delta \Delta G^{‡}=-RTln\left(E\right)$. The modeling is based on the ΔΔG^‡^ values.

#### 4.1.2. Cytochrome P450

The versatile cytochrome P450 (CYP450) family of heme containing redox enzymes hydroxylates a wide range of substrates to generate products of significant medical and industrial importance. Three parental cytochrome P450s, CYP102A1, CYP102A2, and CYP102A3, were used to generate 184 chimeric sequences of cytochrome P450 [6]. A diverse family of thermostable cytochrome P450s was created by recombination of stabilizing fragments. For each variant, the thermostability was analyzed by the measurement of T_50_. T_50_ is the temperature at which 50% of the protein irreversibly denatured after incubation for 10 min. The values of T_50_ range from 39.2 to 64.4 °C.

#### 4.1.3. GLP-2

The GLP-2 dataset involves the potency of 31 alanine variants of the Glucagon-like peptide-2 (GLP-2) with respect to the activation of its receptor [47]. GLP-2 is a short 33-residue peptide whose increase in activity is directly implicated in the control of epithelial growth in the intestine. The value for the corresponding receptor activation for the 31 alanine variants of GLP-2 is defined as the fold increase over basal cyclic adenosine monophosphate (cAMP) production and it ranges from 0.7 to 10.4.

#### 4.1.4. TNF Alpha

The TNF dataset used by Mukai et al. [48] is a collection of 20 mutants and one WT tumor necrosis factor (TNF). TNF is an important cytokine that suppresses carcinogenesis and excludes infectious pathogens to maintain homeostasis. The relative affinity (%K_d_) of TNF to its two receptors, TNFR1 and TNFR2, is computed as a single ratio of log_10_ (R1/R2) which ranges from 0 to 2.87, where R1 and R2 are affinities of TNF to TNFR1 and TNFR2, respectively as measured by IC_50_ assays in ng/mL.

### 4.2. Modeling Approach Based on Single or Multiple Encoding with or without Fast Fourier Transform (FFT)

Our machine learning method named innov’SAR is fully described or commented in previous papers [12,13,22,23,24]. It requires, as input data, only an initial dataset with the primary sequences of protein variants and their corresponding values for a biological activity in order to generate a predictive model. It can be used without knowledge of the 3D structure. It consists of two steps, namely the encoding step and the modeling step.

The encoding step converts the amino acid sequence of the protein into a numerical sequence. This numerical sequence is a required input in order to initiate the modeling step. First, the indices of the AAindex database [27] are used to encode the primary protein sequence into a numerical sequence, where each letter of the amino acid is replaced by a value. This database holds 566 numerical indices representing various physicochemical and biochemical properties for the 20 standard amino acids, and correlations between these indices are also listed. Then, a fast Fourier transformation (FFT) of the encoded sequences from the first step could be performed by innov’SAR to carry out a second layer of encoding. FFT is a digital signal processing technique used to convert numerical signals into an energy versus frequency representation (Equation (1)). After FFT processing, a spectral form of the protein called the protein spectrum is generated. 

A protein spectrum depends on the following equation:(7)$$f_{j}=\overset{N-1}{\underset{k=0}{\sum }}x_{k}\cdot exp\left(\frac{-2i\pi }{N}\cdot j\cdot k\right)$$
where *j* is an index-number of the elementary protein spectrum |*f_j_*|; the numerical sequence includes *N* value(s) denoted *x_k_*, with 0 ≤ *k* ≤ *N* − 1 and *N* ≥ 1; and *i* defines the imaginary number such that *i*² = −1.

At the end of the encoding step, the inputs are used to build models.

The modeling step uses the experimental values of the target activity, in conjunction with encoded sequences in order to identify a predictive model. The model is constructed by the application of regression approaches based on a learning step and a validation step. In this study, the innov’SAR method used a partial least square regression (PLS) algorithm to generate a predictive model of the protein fitness/activity. The learning step leads to the construction of a model. The validation step consists of testing the accuracy of the model in order to check if the learning step was efficient. The root mean squared error (RMSE) and the coefficient of determination (R^2^) are the performance parameters to assess a regression model in the validation step. RMSE values vary between 0 and +∞. The R^2^ value varies between 0 and 1. An accurate regression model has an RMSE close to 0 and a R^2^ close to 1.

One particularity of the innov’SAR method, in the modeling phase, is its ability to evaluate multiple encoding indices to find the best index for the construction of models. The method uses the initial dataset (training set) to construct a predictive model for each encoding index. For each model, the method calculates the value of the performance parameters. The dataset is split into *k* equal portions. The number *k* varies according to the size of the initial dataset. We use a small *k* value if the dataset is small and a bigger *k* value in the opposite case. We use *k-1* portions as the learning dataset and the remaining portion as the validation dataset. The procedure is repeated *k* more times until each portion is used as the testing dataset once. The cross-validation makes it possible to avoid potential overfitting and to optimize some modeling parameters. The method of cross-validation used for this study is the LOOCV, where *k* is equal to the number of sequences. At the end of the modeling step, a ranking is done on the models and their associated encoding indices according to the calculation of the modeling performances. A set of accurate models and their associated encoding indices are identified. This set of models represents the best models to use for the next step described in this paper.

### 4.3. Evaluation of Modeling Performances 

Innov’SAR evaluates the generated models with the values of the RMSE and the R^2^ during the cross-validation stage. 

The cross-validation procedure could be either LOOCV or k-fold, depending on the size of the dataset. Here, it is LOOCV.

The cvRMSE allows the construction and selection of the best models. The predictive ability of these models relies on the R^2^ values. While R^2^ is a measure of the extent of agreement between measured and predicted fitness, cvRMSE represents the extent to which the predictions vary when different training sets are used.

The formulas of these metrics are shown below:(8)$$R^{2}=\frac{\left(\sum_{i=1}^{S}(y_{i}-\bar{y}\right)\left(\hat{y_{i}}-\hat{\bar{y}}\;\right){)}^{2}}{\sum_{i=1}^{S}{\left(y_{i}-\bar{y}\right)}^{2}\sum_{i=1}^{S}{\left(\hat{y_{i}}-\hat{\bar{y}}\right)}^{2}}$$
(9)$$RMSE=\sqrt{\overset{n}{\underset{i=1}{\sum }}\frac{{\left(y-\hat{y}\right)}^{2}}{n}}$$
where, *y_i_* is the measured activity of the *i*^th^ sequence, *ŷ_i_* is the predicted activity of the *i*^th^ sequence, $\bar{y}$ is the average of measured activities, $\hat{\bar{y}}$ is the average of the predicted activities and *n* the number of sequences.

### 4.4. Measure of the Significance of the Improvement between Two Models

The *p*-value associated with the Student’s t-test is used to assess if the difference for the quadratic errors between two models is significant or not.

## 5. Conclusions

Ext_SEQ (extended numerical sequences) as new descriptors were tested on four sets of proteins with different lengths and activities. One very interesting result is that the use of multiple physicochemical descriptors coupled with the implementation of the FFT could significantly improve the results of predictions. This approach is beneficial even when the number of sequences in the train set is limited, as exemplified for GLP-2. The number of indices used has to be evaluated on a case by case basis: Indeed, the concatenation of too many indices could deteriorate the performance. The choice of the descriptor or of the combination of descriptors and/or FFT is dependent on the couple protein/fitness. 

This approach with use of extended numerical sequences (Ext_SEQ) allows, in some cases, a significant improvement of the quality of the models and the predictions of fitness. The implementation of the FFT, taking into account the interactions between residues of amino acids within the protein sequence, leads to the best predictive models. Surprisingly enough, synthetic descriptors such as z-scales or principal components representing various and complementary physicochemical properties are not retained in priority.

Extended numerical sequences appear as more informative descriptors and make it possible to understand the sequences under multiple facets in terms of physicochemical properties. 

The reduction of the variables, through the selection of positions and frequencies, can be a good option to minimize the calculation time, but at the cost of the quality of the prediction.

However, it remains necessary to study the descriptors further, especially those obtained after Fourier transformation. This will be explored in the near future.

We believe that such a methodology, strategically cumulating non-redundant physicochemical information, will be of considerable interest to a large audience of machine learning experts, biotechnologists, molecular biologists, and organic chemists. In particular, it can provide potential users with value added to existing mutant libraries, where screening efforts have so far been unsuccessful in finding improved polypeptide mutants for useful applications.

## Figures and Tables

**Figure 1 ijms-20-05640-f001:**
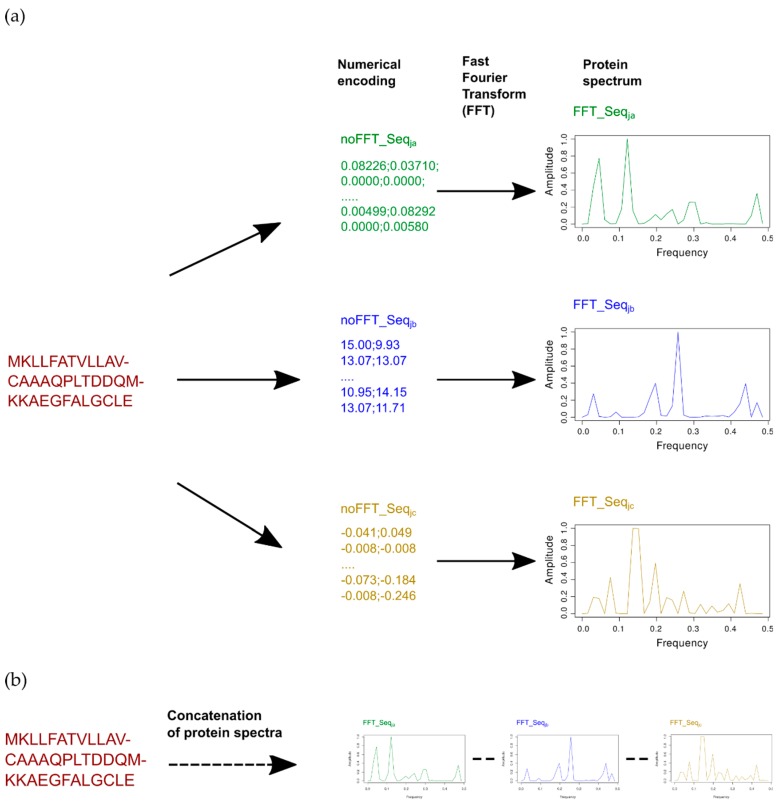
Encoding of sequences with innov’SAR: (**a**) One index encoding: only one index is used and leads to an elementary sequence Ele_(SEQ), and (**b**) Example of an extended sequence of three indexes with FFT: FFT_Seq_ja_—FFT_Seq_jb_—FFT_Seq_jc_: three indices are concatenated and lead to extended numerical sequence (Ext_SEQ). (**c**) A general view of the different phases of innov’SAR with extended sequences. An encoding phase transforms the primary sequences of the initial dataset into protein spectra. The spectra are concatenated to generate extended sequences. The modeling phase uses the extended sequences and the protein activity as a learning dataset in order to construct a regression model. Next, the model performances are evaluated by a comparison between measured and predicted activities.

**Figure 2 ijms-20-05640-f002:**
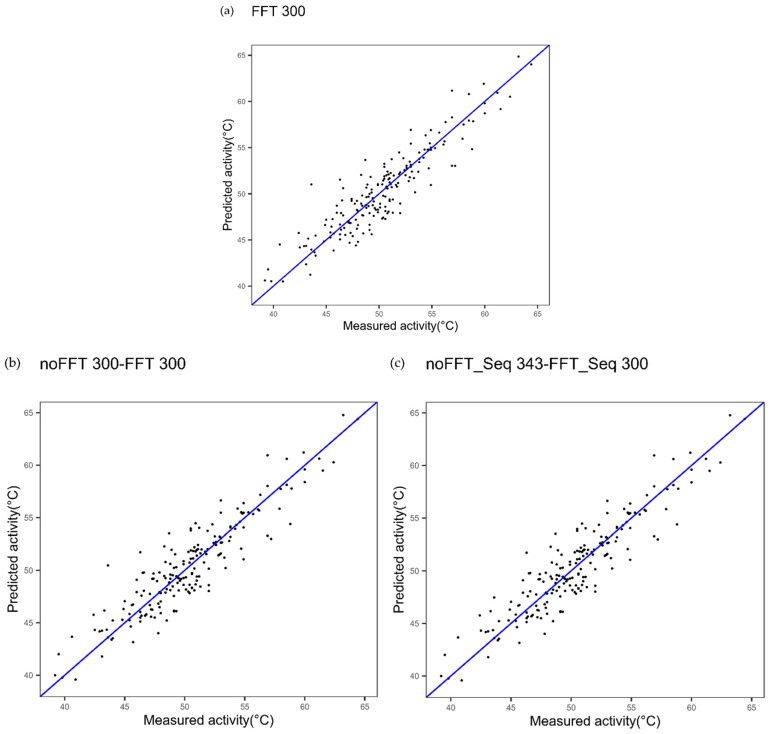
Plot of measured thermostability of cytochrome P450 variants versus predicted thermostability using innov’SAR algorithm with (**a**) one index coupled with Fast Fourier Transform (FFT), (**b**) with the same index coupled and not coupled with FFT and, (**c**) two indices, one coupled with FFT and the other not. The number of the index in the amino acid index (AAindex) database is indicated at the top of the plot: “300”.

**Figure 3 ijms-20-05640-f003:**
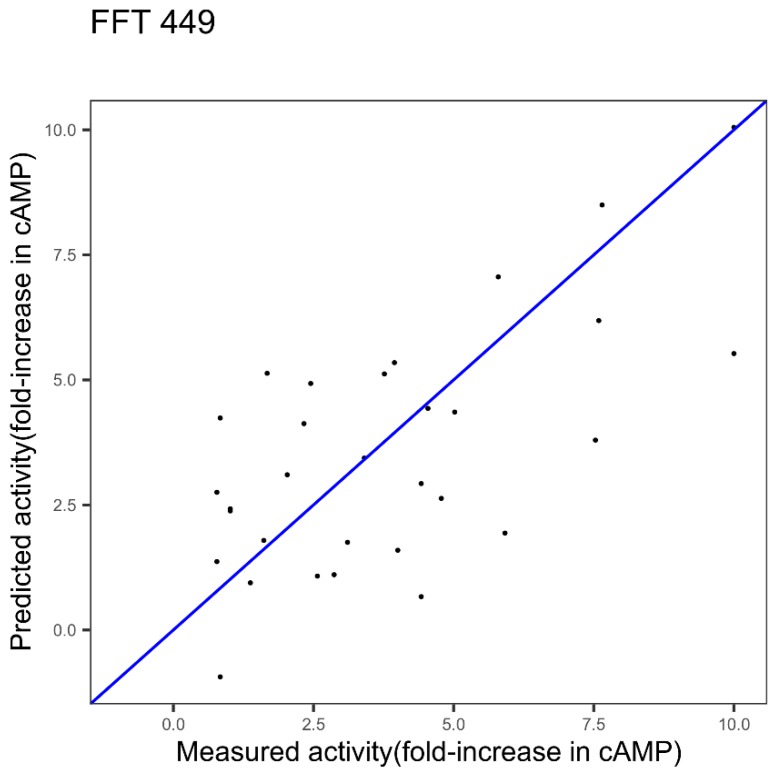
Plot of measured potency (fold-increase in cyclic adenosine monophosphate, cAMP, concentration) of Glucagon-like peptide-2 (GLP-2) variants versus predicted potency using innov’SAR algorithm with one index coupled with Fast Fourier Transform (FFT).

**Figure 4 ijms-20-05640-f004:**
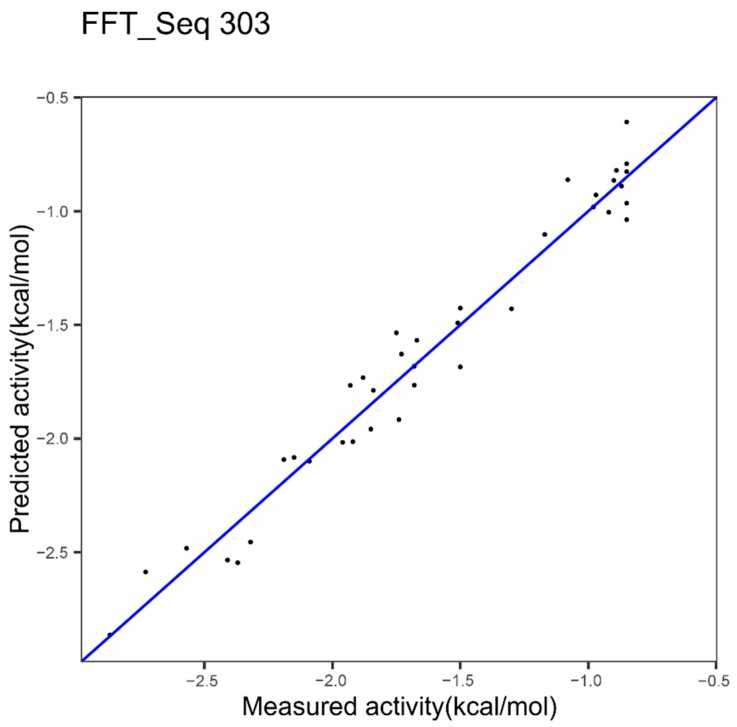
Plot of measured ΔΔG^‡^ of epoxide hydrolase variants versus predicted ΔΔG^‡^ using innov’SAR algorithm with one index coupled with Fast Fourier Transform (FFT).

**Figure 5 ijms-20-05640-f005:**
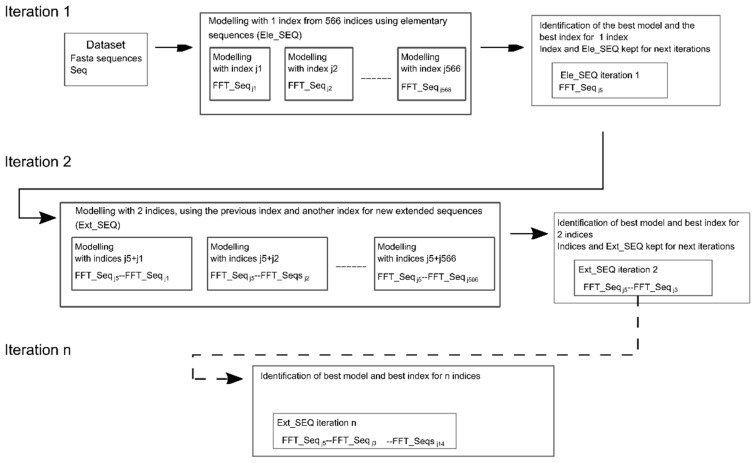
Workflow of the iterative process for successive concatenation. Each round uses the indices from the previous iteration as a base of extended sequence (Ext_SEQ) and determines the best index to keep for the current round by evaluation of the modeling performances.

**Figure 6 ijms-20-05640-f006:**
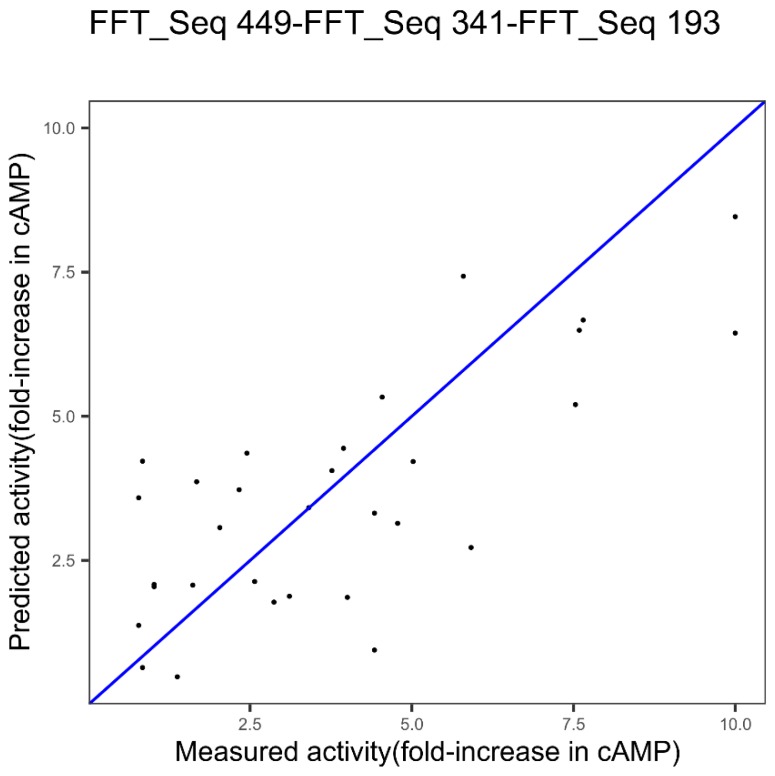
Plot of measured potency (fold-increase in cyclic adenosine monophosphate, cAMP, concentration) of Glucagon-like peptide-2 (GLP-2) variants versus predicted potency using innov’SAR algorithm with three indices coupled with Fast Fourier Transform (FFT).

**Figure 7 ijms-20-05640-f007:**
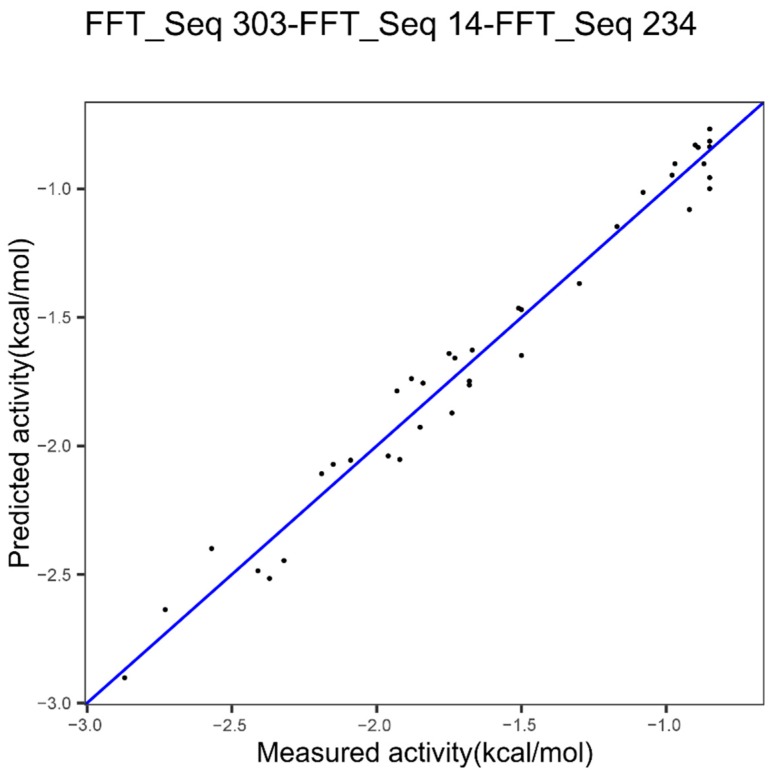
Plot of measured ΔΔG^‡^ of epoxide hydrolase variants versus predicted ΔΔG^‡^ using innov’SAR algorithm with three indices coupled with Fast Fourier Transform (FFT).

**Figure 8 ijms-20-05640-f008:**
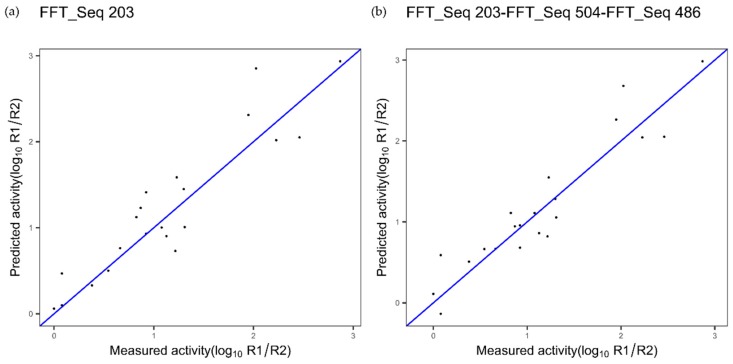
Plot of measured affinity versus predicted affinity of tumor necrosis factor (TNF) variants using (**a**) a single index or (**b**) three indices coupled with Fast Fourier Transform (FFT).

**Figure 9 ijms-20-05640-f009:**
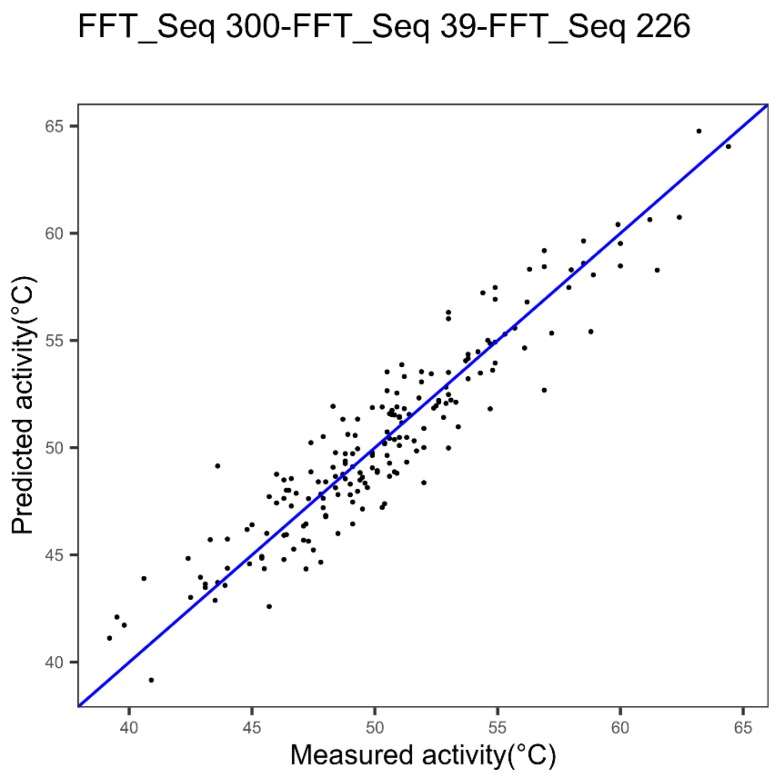
Plot of measured versus predicted thermostability of cytochrome P450 variants using three indices coupled with Fast Fourier Transform (FFT).

**Figure 10 ijms-20-05640-f010:**
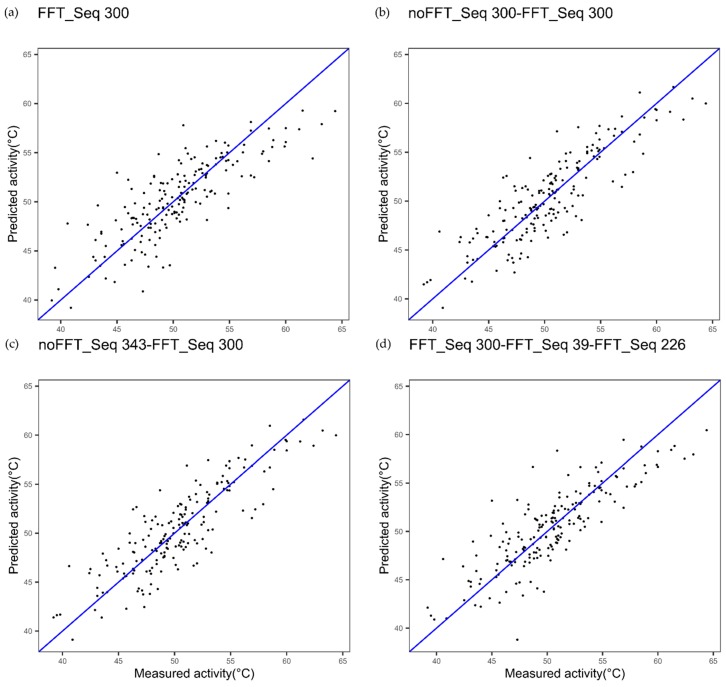
Plot of measured versus predicted thermostability of cytochrome P450 variants using (**a**) a single index (300) coupled with Fast Fourier Transform (FFT) and a selection of 20% frequencies, (**b**) a single index (300) twice, once without FFT and another time coupled with FFT and by selection of 20% frequencies, (**c**) using two indices, one index without FFT (343) and one index coupled with FFT (300) and by selection of 20% frequencies, and (**d**) using innov’SAR algorithm with three indices (300, 39, 226) coupled with FFT and by selection of 20% frequencies.

**Table 1 ijms-20-05640-t001:** GLP-2: Top 10 extended sequences selected using a combinatorial approach applied on the top 10 best indices.

Index	cvRMSE	cvR^2^
440 350 44	1.99	0.47
440 350	1.99	0.47
440 350 233	1.99	0.47
440 44	2.06	0.37
44 233	2.09	0.37
449 350	2.10	0.43
449 350 233	2.10	0.43
449 350 44	2.10	0.43
449	2.11	0.42
449 233	2.11	0.42

**Table 2 ijms-20-05640-t002:** Epoxide hydrolase: Top 10 extended sequences selected using a combinatorial applied to the top 10 best indices.

Index	cvRMSE	cvR^2^
161 178 516	0.1051	0.9685
254 178 516	0.1051	0.9685
232 161 508	0.1123	0.9640
232 254 508	0.1123	0.9640
161 508	0.1146	0.9629
254 508	0.1146	0.9629
161 254 508	0.1150	0.9624
303 508	0.1161	0.9624
303 161 508	0.1170	0.9615
303 254 508	0.1170	0.9615

**Table 3 ijms-20-05640-t003:** Identification of the protein feature families involved in modeling.

Dataset	Index 1 and Protein Feature	Index 2 and Protein Feature	Index 3 and Protein Feature
GLP-2	Index 449, other properties family	Index 341, alpha and turn propensities family	Index 193, composition family
Epoxide hydrolase	Index 303, other properties family	Index 14, hydrophobicity	Index 234, beta propensity
TNF alpha	Index 203, composition	Index 504, NA	Index 486, NA
Cytochrome P450	Index 300, alpha and turn propensities	Index 39, beta propensity	Index 226 beta propensity

**Table 4 ijms-20-05640-t004:** Summary of the performance metrics values obtained on the four datasets and for the different kinds of experiments carried out. “-“ indicates that the corresponding model is used as a reference for calculation of the *p*-value.

Dataset	cvRMSE	cvR^2^	*p*-Value (Error² with Paired Student Test)
**Cytochrome P450**			
Cytochrome P450 FFT one index Figure 2a	1.91	0.83	-
Cytochrome P450 multi indices noFFT_FFT Figure 2c	1.85	0.84	0.4123
Cytochrome P450 one index noFFT + FFT Figure 2b	1.89	0.83	0.7781
Cytochrome P450 3 indices FFT Figure 9	1.63	0.88	0.0020
**Cytochrome P450 - selection of 20% frequencies**			
Cytochrome P450 1 index FFT selection 20% Figure 10a	2.68	0.66	-
Cytochrome P450 two indices noFFT_FFT selection 20% Figure 10c	2.39	0.74	0.0767
Cytochrome P450 one index noFFT + FFT selection 20% (Figure 10b)	2.38	0.74	0.0600
Cytochrome P450 3 indices FFT selection 20% Figure 10d	2.52	0.7	0.0826
**GLP-2**			
GLP-2 1 index FFT Figure 3	2.11	0.42	-
GLP-2 3 indices FFT Figure 6	1.75	0.55	0.0078
GLP-2 3 indices best in Table 1	1.99	0.47	0.5309141
**Epoxide Hydrolase**			
Epoxide Hydrolase 1 index FFT 303 Figure 4	0.12	0.96	-
Epoxide Hydrolase best in Table 2	0.1051	0.9685	0.4322954
Epoxide Hydrolase 3 indices FFT Figure 7	0.094	0.9747	0.3435683
**TNF Alpha**			
TNF 1 index 203 Figure 8a	0.32	0.85	-
TNF 3 indices FFT Figure 8b	0.28	0.88	0.1749

**Table 5 ijms-20-05640-t005:** Correspondence of the index number and the name and reference of the index in the AAindex database [27].

Index Number	Index Name	Applied on Dataset
39	Normalized frequency of beta-sheet [30]	CYP450
226	Normalized frequency of beta-sheet from CF [31]	CYP450
300	Average relative fractional occurrence in A0(i) [32]	CYP450
343	Information measure for extended [33]	CYP450
450	Hydropathy scale based on self-information values in the two-state model (25% accessibility) [34]	CYP450
14	Transfer free energy to surface [35]	Epoxide H.
161	Normalized frequency of beta-sheet, with weights [36]	Epoxide H.
178	Retention coefficient in HPLC, pH7.4 [37]	Epoxide H.
232	Normalized frequency of beta-sheet in all-beta class [31]	Epoxide H.
254	Relative frequency in beta-sheet [38]	Epoxide H.
303	Average relative fractional occurrence in EL(i) [32]	Epoxide H.
508	Linker propensity from helical (annotated by DSSP) dataset [39]	Epoxide H.
516	Hydrostatic pressure asymmetry index, PAI [40]	Epoxide H.
44	Normalized frequency of C-terminal non-helical region [30]	GLP-2
193	AA composition of mt-proteins from animal [41]	GLP-2
233	Normalized frequency of beta-sheet in alpha + beta class [39]	GLP-2
341	Information measure for middle helix [33]	GLP-2
350	Information measure for coil [33]	GLP-2
440	Distribution of amino acid residues in the 18 non-redundant families of thermophilic proteins [42]	GLP-2
449	Hydropathy scale based on self-information values in the two-state model (20% accessibility) [34]	GLP-2
203	AA composition of CYT2 of single-spanning proteins [43]	TNF
297	Average reduced distance for C-alpha [44]	TNF
486	Electron-ion interaction potential values [17]	TNF
504	Linker propensity from three-linker dataset [39]	TNF
523	Apparent partition energies calculated from Chothia index [45]	TNF

**Table 6 ijms-20-05640-t006:** Comparison of the calculation time for the sequential (A) and combinatorial (B) approach.

Dataset	Calculation Time for Method A (min)	Calculation Time for Method B (min)
GLP-2	5	2
Epoxide hydrolase	60	9
TNF alpha	15	3
Cytochrome P450	120	ND

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
