# Peer review of "Novel Descriptors and Digital Signal Processing- Based Method for Protein Sequence Activity Relationship Study"

_ijms, 2019, doi:10.3390/ijms20225640_

Round 1

Reviewer 1 Report

This manuscript “Novel Descriptors and Digital Signal Processing based method for Protein Sequence Activity Relationship Study” by  Fontaine et al. proposes a method to select the most significant descriptors describing amino acid sequences of proteins to correlate the sequences with the “fitness” of proteins.  The rationale is understandable and acceptable; however, there are still some minor issues to be addressed:

1. The authors used four datasets: Epoxide hydrolase, P450, GLP-2 and TNF alpha, but they tried to predict four different properties of those proteins. The rationale why the authors predicted different properties and how they decided which properties should be predicted was not disclosed.
2. How did the authors choose features to predict? Did they exhaustedly try all combinations?
3. In the discussion, please try to connect the input features and the output properties. Why it works is the most interesting topic and worth to know for a reader.

Reviewer 2 Report

   The authors have presented a new way of considering descriptors from the Amino Acids index database for modeling and presenting the fitness value of a polypeptide chain. The authors also show that the use of multiple physicochemical descriptors coupled with the implementation of the FFT could lead to very significant improvement in the quality of models and predictions.

   However, I have considered that the authors’ explanations in the manuscript are insufficient in some respects, as described in major and minor corrections below, for readers studying in different research fields to understand more easily the content of this manuscript.

Major corrections

Figure 1: First of all, the authors should explain for readers to understand more easily, what index was used in three one index encodings, and how the amino acid sequence composed of 36 amino acid residues was transformed into eight decimals. In addition, vertical and horizontal axes of the figures after FFT_Seqj should be specified and dot plots, which are finally obtained, should be presented in Figure 1. The authors should specify the unit on vertical and horizontal axes in the Figures 2 (oC?), 4 (kJ/mole?), 7 (kJ/mole?), 8 (fold increase in affinity?) and 9 (oC?), because it is difficult to understand the figures. 

Minor corrections

Figure 2a should be Figure 2. Figure 8a should be Figure 8. In addition, (a) and (b) should be added at the top of the respective plots. Values for cvRMSE and cvR2 should be written after the sentence of line 276, as “as 0.96 and 0.12 for cvRMSE and cvR2, respectively”. Line 280: “Table” should be “Table 2”. The authors should explain the reason why the number of dots in Figure 8 is considerably few as around 20 than 129 for TNF. Figure 10a should be Figure 10. In addition, (a), (b), (c) an (d) should be added at the top of the respective plots. Which is correct, 2.39 on Line 451 and 2.9 in Table 4? “Digital Signal Processing (DSP)” on Line 63 and “Digital Signal Processing” on Line 68 should be “DSP”, because “Digital Signal Processing (DSP)” is previously described on Line 61. “Fast Fourier Transform” on Line 69 should be “Fast Fourier Transform (FFT). “Leave-One-Out Cross-Validation” should be written before “LOOCV” on Line 177, which is appeared for the first time in the manuscript. “Leave-One-Out Cross-Validation” on Line 670 should be deleted. “the root mean squared error (RSME)” on Line 676 should be revised as RSME, because the explanation of RMSE is previously described on Line 657. Line 356: “work26” should be “work [26]”. Line 357: “Innov’sar” should be “Innov’SAR”. Line 379: Ele_seq should be Ele_Seq. Line 403: “4. selection of variables inside the ext_seq” should be “2.4. Selection of variables inside the Ext_Seq”. Single space should be deleted from between “determination” and “,” on Line 74. Single space should be inserted at study[25] on Line 242, study[26] on Line 270, 3indices on Line 316, Table4 on Line 462 and Tomii et al.[32] on Line 562, as study [25], study [26], 3 indices, Table 4 and Tomii et al. [36], respectively. Double space should be revised to single space at many locations, as follows. Line 54: between “present.” and “Other”, Line 74: between “parameter” and “determination”, between “by” and “glucose”, Line 75: between“values” and “have”, Line 178: between “cvRMSE” and “rises”, Line 179: between “index:” and “thus”, Line 242: between “[25]” and “the”, Line 243: between “shown” and “as” Line 245: between “[25].” and “cvR2”, Line 338: between “best” and “index”, Line 396: between “indices:” and “with”, Line 456: between “selection,” and “of”, Line 457: between “FFT (300)” and “and” and between “selection” and “of”, Line 459: between “selection” and “of”, Line 492: between “indices.” and “Student’s”, Line 530: between “on” and “fitness”, Line 548: between “following:” and “syntheses”, Line 565: between “performances.” and “In”, Line 595: between “greater.” and “Table”, and Line 615: between “T50.” and “T50”.

Round 2

Reviewer 2 Report

The authors have revised the previous manuscript following my questions and comments.